# Genetic Structure of an East Asian Minnow (*Toxabramis houdemeri*) in Southern China, with Implications for Conservation

**DOI:** 10.3390/biology11111641

**Published:** 2022-11-09

**Authors:** Weitao Chen, Yuefei Li, Xingwei Cai, Denggao Xiang, Shang Gao, Ce Li, Chun Lan, Shuli Zhu, Jiping Yang, Xinhui Li, Jie Li

**Affiliations:** 1Pearl River Fisheries Research Institute, Chinese Academy of Fishery Sciences, Guangzhou 510380, China; 2Key Laboratory of Aquatic Animal Immune Technology of Guangdong Province, Guangzhou 510380, China; 3Guangzhou Scientific Observing and Experimental Station of National Fisheries Resources and Environment, Guangzhou 510380, China; 4Hainan Academy of Ocean and Fisheries Sciences, Haikou 571126, China; 5College of Marine Ecology and Environment, Shanghai Ocean University, Shanghai 201306, China; 6Aquatic Technology Extension Station of Du’an County, Hechi 530700, China

**Keywords:** genetic structure, *Toxabramis houdemeri*, southern China, river rearrangements, conservation implications

## Abstract

**Simple Summary:**

This study presents the first comprehensive view of the genetic structure of a widespread cyprinid, *Toxabramis houdemeri,* based on large-scale geographic sampling and mitochondrial and nuclear markers. Genetic endemism with clear geographic boundaries was formed in *T. houdemeri* populations due to river landscape changes, biogeographic barriers, and the species’ dispersal potential. Late Pleistocene demographic expansion had occurred in *T. houdemeri* populations. This study could help improve the monitoring and protection of this species.

**Abstract:**

River dynamics have been hypothesized to substantially influence the genetic structure of freshwater fish taxa. Southern China harbors abundant independent river systems, which have undergone historical rearrangements. This river system is thus an excellent model with which to test the abovementioned hypothesis. In this study, a cyprinid widespread in many independent rivers in southern China, *Toxabramis houdemeri*, was chosen as an exemplar species with which to explore the effects of river configuration changes on spatial genetic structure using mitochondrial and nuclear markers. The results indicated that the *T. houdemeri* populations fell into four mitochondrial haplotype groups, each genetically endemic to a single river or two adjacent river systems. The mitochondrial haplotype network recovered a clear genetic boundary between Hainan Island populations and mainland populations. Notable genetic differentiation was observed within populations from distinct river systems in both mitochondrial and nuclear loci. River system separation, mountain barriers, and mobility were the key factors shaping the genetic structure of *T. houdemeri* populations. Late Pleistocene divergence and historical immigration were identified within the four mitochondrial haplotype groups, indicating that river rearrangements triggered by the Late Pleistocene glacial cycles were important drivers of the complex genetic structure and demographic history of *T. houdemeri*. Historical demographics suggested that *T.*
*houdemeri* populations expanded during the Late Pleistocene. The present study has important consequences for the management and conservation of *T. houdemeri*.

## 1. Introduction

For effectively managed conservation, it is critical to understand the spatial configurations of genetic diversity and to excavate genetic endemism (unique genetic resources in a particular geographic region) within species, particularly in the case of widespread species. In particular, since widely distributed populations of freshwater fish may frequently be isolated by mountains or salinity [1], gene flow is likely to be interrupted, and genetic differentiation and/or endemism may occur within populations [2,3,4,5,6,7]. In addition, it has been argued that river rearrangements triggered by glacial cycles may have both blocked gene exchange by isolating previously connected rivers and promoted population dispersal by connecting previously isolated rivers [1,2,3,4,5,6,7,8]. River rearrangements have historically been more common in rivers near the sea [4,7]. For example, the Last Glacial Maximum led to sea-level retrogression, thus promoting the confluence of rivers previously isolated by salt water and allowing population migrations between previously isolated rivers [5,9]. Conversely, during interglacial periods, sea-level increases disrupted river connections, leading to population differentiation or even generating genetic endemism in certain isolated rivers [7,10]. Therefore, changes in riverine landscape structure both present and past may strongly affect patterns of genetic diversity within freshwater fish populations.

In addition to the large Pearl River, many independent coastal rivers are distributed in southern China, such as the Moyang River and the Jian River on the mainland and the Nandu River, the Changhua River, and the Wanquan River on Hainan Island (Figure 1). These rivers are currently separated but spatially proximate and discharge into the South China Sea. Previous geological reports have shown that the South China Sea has been affected by several worldwide glacial cycles since the Pleistocene [11,12], which have influenced the landscape configurations of rivers in this region. It is thus likely that the genetic structures of the fish species residing in these rivers have been shaped by periodic riverine connectivity and disruption.

Several genetic studies of freshwater fish in southern China have detected high levels of population differentiation [2,3,7,8,9,10,13]. For example, Yang and He [2] found that river landscape changes induced by the Pleistocene glacial cycles were significant drivers of the high levels of population divergence within *Hemibagrus guttatus* populations in the Pearl River, Hanjiang River, and Jiulongjiang River [2]. Similarly, Chen et al. [7] found marked genetic differentiation and endemism within populations of the black Amur bream (*Megalobrama terminalis*) in the Pearl River, the Moyang River, and the Wanquan River, and the authors argued that river rearrangements mediated by glacial cycles shaped this remarkable genetic pattern [7]. However, most previous genetic reports have focused on only a limited number of independent rivers in southern China. Therefore, a comprehensive examination of the effects of river landscape on the genetic structures of fish taxa is necessary.

*Toxabramis houdemeri* (Pellegrin, 1932), a small cyprinid (length at maturity < 14.7 cm; www.fishbase.cn, accessed on 28 September 2022) found in numerous rivers in southern China, is a good exemplar species for explorations of the influences of river landscape alternations on the genetic patterns of freshwater fish taxa. *T. houdemeri* is a mesopelagic fish that is considered a poor disperser due to its small body size [14,15]. Given the increasing threats posed by human interference and/or environmental degradation to freshwater biodiversity worldwide [16,17,18], it is crucial to protect current stocks of *T. houdemeri*. Resolving the spatial genetic structure of *T. houdemeri* and revealing the processes influencing this structure will provide a solid theoretical basis for the development of effective conservation strategies.

In this study, we sampled *T. houdemeri* populations from several independent coastal rivers in southern China and investigated population genetic structure using both mitochondrial and nuclear loci. We aimed (i) to investigate the spatial genetic structure of *T. houdemeri* populations and to identify genetic endemism in different rivers; (ii) to characterize population demographics and determine potential influencing factors; and (iii) to provide a relevant theoretical basis for the development of future conservation strategies.

## 2. Methods

### 2.1. Sampling

During field surveys between 2014 and 2022, we sampled 529 specimens of *T. houdemeri* from 26 localities in the rivers of southern China, including the Pearl River and eight independent coastal rivers (Appendix A; Figure 1). A small piece of fin or muscle was clipped from each specimen and preserved in 95% ethanol.

Total genomic DNA was extracted from fin or muscle tissue samples using the Axygen DNA Extraction Kit (Axygen Scientific, Union City, CA, USA), following the manufacturer’s instructions. Two mitochondrial fragments were amplified: the partial mitochondrial cytochrome *b* gene (*Cytb*) and the control region (CR) using universal primers L14724 and H15915 [19] and DL1 and DH2 [20]. We also amplified and sequenced a nuclear locus (recombination activating gene 2, *RAG2*) using the primers *RAG2*-f2a and *RAG2*-R6a [21]. All selected markers were amplified and sequenced as previously described [19,20,21]. The amplification conditions were identical to those described by Chen et al. (2020) [7].

### 2.2. Sequence Analyses and Haplotype Network

The nucleotide sequences were initially aligned using MUSCLE [22] and manually optimized using MEGA X [23]. The mitochondrial fragments *Cytb* and CR were concatenated into one mitochondrial locus (MCR) for subsequent analyses. For *RAG2*, alleles were unphased using the PHASE algorithm as implemented in DnaSP v5.0 [24] with default settings. Identical MCR haplotypes and phased nuclear gene alleles were collapsed using DnaSP v5.0.

Due to low intraspecific sequence divergence and low resolution in the phylogenetic topology, we did not build phylogenetic trees for the *T. houdemeri* populations. Instead, we used PopART v1.7.2 [25] to infer mitochondrial haplotype and nuclear allele relationships between nine independent rivers. Haplotype networks can be used to infer precise relationships between closely related populations [25]. MCR sequences were directly utilized for haplotype network construction. The longest non-recombining region generated in IMGC [26] for *RAG2* was used to build the allele network.

### 2.3. Molecular Diversity and Genetic Structure

To characterize the genetic diversity of the *T. houdemeri* populations, we calculated the number of haplotypes (h), haplotype diversity (H*d*), and nucleotide diversity (*π*) using DnaSP v5.0. Genetic differentiation (the fixation index ϕ_ST_ [27]) was assessed in ARLEQUIN v3.5 [28] by calculating pairwise ϕ_ST_ values between populations. To test whether the *T. houdemeri* populations fit an isolation-by-distance pattern, a Mantel test was performed in ARLEQUIN v3.5. The geographical distance (in kilometers) was roughly measured as the straight-line distance using Google Earth v.4.3. Hierarchical and non-hierarchical population structures based on both types of loci were examined using analyses of molecular variance (AMOVAs) in ARLEQUIN v3.5. First, we estimated the overall differentiation of the complete data set without partitions. Second, we calculated the partitioning of genetic variation between the nine independent rivers and four defined genetic groups (see Section 3). All calculations implemented in ARLEQUIN v3.5 were performed with 1000 permutations.

### 2.4. Divergence Time Estimate

We used pairwise net average sequence distances between species to estimate the approximate divergence times between the species in the four observed genetic groups. We calculated the net average sequence distance between genetic group pairs in MEGA X to estimate approximate divergence times. The average net Kimura-2-Parameter (K2P [29]) distance for the MCR compared to *Cytb* alone was 0.756. The estimated mean substitution rates for MCR (0.756% and 1.513%) were obtained by multiplying the *Cytb* rate by the K2P ratio for *Cytb* alone (0.756). The estimated substitution rates for MCR were used to measure the approximate divergence times between group pairs.

### 2.5. Demographic History and Historical Gene Flow

Three approaches were used to infer the demographic histories of the *T. houdemeri* populations. First, to test for departures from a constant population size, the summary statistics Tajima’s *D* [30] and Fu’s *F_S_* [31] were estimated. Significance was assessed based on 1000 simulated samples. Second, pairwise mismatch distributions [32] were used to infer the demographic history and were performed using ARLEQUIN v3.5 and DnaSP v5.10. Third, we examined historical demographics using coalescent-based extended Bayesian skyline plots (EBSPs) [33]. EBSPs were performed using both mitochondrial and nuclear loci in BEAST v1.8.1 [34]. Mitochondrial *Cytb* and CR were separated in this analysis. Before conducting EBSPs, we chose the optimal model of nucleotide substitution for each locus (Appendix A) using MrModeltest v2.3 [35]. The range of substitution rates (1.0–2.0% per million years, Myr) adopted for the *Cytb* gene in this study was selected based on commonly acknowledged rates in cyprinid fish [36,37,38,39]. The substitution rates for CR and *RAG2* were measured as a function of the *Cytb* evolutionary rate. EBSPs were run with a strict clock model and 30 million generations. Finally, TRACER v1.6 [40] was employed to assess the stationarity of each run by analyzing the effective sizes of all parameters.

To assess the direction and magnitude of historical gene flow between the four haplotype groups, MIGRATE v4.5 [41] was utilized to calculate the historical migration rate. Three independent runs using the maximum likelihood strategy were executed. One long chain (20,000 genealogies sampled) and 12 short chains (5000 genealogies sampled) with temperatures of 1.0, 1.5, 3.0, and 100,000.0 and a burn-in of 10,000 genealogies per chain were implemented in the MCMC searches. We set the initial uniform priors *Θ* and *M* to 0.0–0.1 and 0.0–25,000.0, respectively.

## 3. Results

### 3.1. Sequence Statistics

Fragments of *Cytb* (1105 bp) and the CR (772 bp) were successfully sequenced from 527 samples. In total, we obtained 527 MCR sequences (1877 bp), with 147 variable sites and 106 haplotypes. We yielded *RAG2* sequences (1201 bp) from 509 specimens, with 74 variable sites and 168 alleles. All novel sequences obtained in this study have been deposited in GenBank (Appendix A).

### 3.2. Haplotype and Allele Networks

The mitochondrial network revealed that the Pearl River populations and the Moyang River population represented different private haplotype groups. In addition, the populations from the Jian River and the Lian River shared two main haplotypes, while the remaining haplotypes found in the two rivers were not shared (Figure 2a). There was a clear genetic delineation between the Hainan Island populations and the mainland populations. However, several haplotypes were shared among the populations from the five Hainan Island rivers. Based on the mitochondrial haplotype network, we defined four independent genetic groups: the Pearl River populations, the Moyang populations, the Jian and Lian populations, and the Hainan Island populations (groups G1, G2, G3, and G4, respectively). In the nuclear networks, many alleles were shared across the nine rivers, and a handful of alleles were unique in some rivers (Figure 2b).

### 3.3. Genetic Structure

The nonhierarchical AMOVAs identified statistically significant differences across populations, as indicated by high global ϕ_ST_ values, for both loci (ϕ_CT_ = 0.539, *p* < 0.001 for MCR and ϕ_CT_ = 0.414, *p* < 0.001 for *RAG2*; Table 2). The estimated ϕ_CT_ value for the nine independent rivers, calculated using hierarchical AMOVAs, also recovered significant genetic differentiation (ϕ_CT_ = 0.253, *p* < 0.001 for MCR and ϕ_CT_ = 0.322, *p* < 0.001 for *RAG2*; Table 2). Hierarchical AMOVAs identified similar levels of significant genetic differentiation between the four defined generic groups for both loci (ϕ_CT_ = 0.379, *p* < 0.001 in MCR and ϕ_CT_ = 0.377, *p* < 0.001 in *RAG2*; Table 2). Within the Pearl River, nonhierarchical AMOVAs revealed relatively high levels of significant genetic differentiation between populations (ϕ_CT_ = 0.371, *p* < 0.001 in MCR and ϕ_CT_ = 0.240, *p* < 0.001 in *RAG2*; Appendix A). In contrast, there was relatively low genetic differentiation between the Hainan Island populations (ϕ_CT_ = 0.149, *p* < 0.001 in MCR and ϕ_CT_ = 0.093, *p* < 0.001 in *RAG2*; Appendix A).

The range of pairwise ϕ_ST_ values for MCR was 0.000–0.965, and 251 out of 276 ϕ_ST_ values were significant (Appendix A). The range of pairwise ϕ_ST_ values for *RAG2* was 0.000–0.925, and 257 out of 276 ϕ_ST_ values were significant (Appendix A). Additionally, for both of these loci, pairwise ϕ_ST_ values were relatively high among the Pearl River populations but relatively low among the Hainan Island populations (Appendix A). Mantel tests identified a significant correlation between geographical and genetic differentiation in MCR (*r* = 0.259, *p* < 0.001) and *RAG2* (*r* = 0.451, *p* < 0.001).

### 3.4. Molecular Diversity Indexes

The H*d* and π values of each population, overall samples, and the defined genetic groups were showed in Table 1. For MCR, the global haplotype (H*d*) and nucleotide diversity (π) values were 0.948 and 0.0044, respectively. The highest H*d* and π values within the four defined groups were detected in G1. With respect to *RGA2*, the overall H*d* and π values were 0.915 and 0.0036, respectively. The highest H*d* and π values within the four genetic groups were examined in G3.

### 3.5. Demographic Analyses and Gene Flow

Both Tajima’s *D* and Fu’s *F_S_* for the overall populations were significantly negative (Appendix A). Additionally, mismatch analyses revealed multimodal distribution of pairwise differences for both loci (Appendix A). The model of sudden demographic expansion was not rejected by the generalized least square procedure (SSD = 0.005, *p* = 0.355 for MCR and SSD = 0.003, *p* = 0.690 for *RAG2*) or by the raggedness index of the distribution (Rag = 0.008, *p* = 0.421 for MCR and Rag = 0.013, *p* = 0.740 for *RAG2*) (Appendix A). Lastly, EBSP analyses using the substitution rates of 1% and 2% for *Cytb* detected a population expansion from ~0.06 million years ago (Ma) to 0.120 Ma (Figure 3).

Using substitution rates of 0.756% and 1.513% per Myr for MCR, estimates of historical migration between the four groups showed symmetrical magnitude and direction between G1 and G2, between G1 and G3, and between G2 and G3. Additionally, asymmetrical bias in historical gene flow was detected between G1 and G4, between G2 and G4, and between G3 and G4. The median estimates of the number of effective migrants per generation originating from G1 and traveling into G4 was 0.175 (0.756%)/0.349 (1.513%), while the median number of migrants per generation traveling in the opposite direction was estimated at 0.044 (0.756%)/0.089 (1.513%). Estimates of migration between G2 and G4, as well as between G3 and G4, showed similar patterns (Figure 4a).

### 3.6. Divergence Time Estimates

Based on average sequence distances, the divergence time range for the four defined groups was 0.119–0.192 Ma based on a substitution rate of 1.513% and was 0.238–0.384 Ma based on a substitution rate of 0.756% (Figure 4b).

## 4. Discussion

### 4.1. Genetic Structure of T. houdemeri

Four mitochondrial haplotype groups with rigid geography were observed in the *T. houdemeri* populations, suggesting long-term geographical separation. Divergence time estimates suggested that the four defined groups split at ~0.119–0.384 Ma, which indicated isolation during the Late Pleistocene. In addition, the historical gene flow between the four genetic groups indicated that, at some point in history, these independent rivers were connected, allowing overlapping dispersal and gene flow. River rearrangements during the Late Pleistocene glacial cycles [42] appear to have been a key factor that shaped the complex evolutionary histories. During the glacial period, the reduced sea levels in the South China Sea generated wide continental shelves and land bridges and also connected some of the rivers in this region, which may have accelerated gene flow between *T. houdemeri* populations. At the end of the glacial period, the rising sea levels [11,12,43] isolated Hainan Island, the Moyang River, the Jian and Lian rivers, and the Pearl River, hindering gene exchange between *T. houdemeri* populations. Similar population structure patterns have previously been documented in many other freshwater fish species in southern China, including *Culter recurviceps* [8], *Megalobrama terminalis* [7], *Rhinogobius duospilus* [10], and *Schistura fasciolata* [3].

Significant differentiation (ϕ_CT_ = 0.253 for MCR and ϕ_CT_ = 0.322 for *RAG2*) between the nine independent river populations indicated that riverine isolation might have influenced the genetic architecture of *T. houdemeri* populations. Nowadays, these nine rivers are independent and unconnected, divided by both terrestrial and/or marine areas. Consequently, the *T. houdemeri* populations in these rivers are each restricted to single rivers, resulting in obvious genetic differentiation.

Interestingly, endemic mitochondrial haplotypes were discovered in the Moyang River even though the Moyang River and various branches of the Pearl River are isolated only by tens of kilometers (Figure 1). Furthermore, high levels of genetic differentiation between the Moyang River populations and the Pearl River populations were recovered by AMOVA and ϕ_ST_ calculations in both loci. A similar pattern was observed in *M. terminalis* [7]. The Yunkai Mountains, an important biogeographical barrier in southern China, have played a noteworthy role in shaping the genetic structures of the *T. houdemeri* populations. The Pearl River and the Moyang River are located on the flanks of the Yunkai Mountains, and this geographic placement has potentially hindered gene exchange between the two rivers. Consistently, it has been proposed that during the Pleistocene, the Yunkai Mountains facilitated the differentiation of several freshwater species and populations [7,9,44,45]. The unique mitochondrial haplotypes identified in the Jian River and the Lian River may also have arisen due to population separations associated with the Yunkai Mountains. The phenomenon of haplotype sharing occurring between the Jian River population and the Lian River population might be due to the spatial proximity of the two rivers (Figure 1).

As expected, the mitochondrial network revealed a clear genetic boundary between the Hainan Island populations and the mainland populations (Figure 2a), likely due to the isolating effects of the Qiongzhou strait. This strait is an acknowledged phylogeographic break that has been identified as a key force driving genetic differentiation between Hainan Island and Mainland China freshwater fish [7,8,13,44,46], such as *Micronemacheilus pulcher* [13], *Garra orientalis* [9], *Megalobrama terminalis* [7], and *Culter recurviceps* [8].

Unexpectedly, AMOVA and ϕ_ST_ estimates of genetic differentiation between the Pearl River populations based on the mitochondrial and nuclear loci were relatively high. This result was incongruent with the outcome in *M. terminalis* (subfamily Cultrinae) in the Pearl River, which found no population structure among *M. terminalis* populations [7]. Furthermore, the genetic differentiation between *T. houdemeri* populations in the Pearl River was also greater than that between *C. recurviceps* populations in the Pearl River [8]. This discrepancy may be due to the limited dispersal potential of *T. houdemeri,* considering *T. houdemeri* is a small fish (<14.7 cm; www.fishbase.cn: accessed on 10 September 2022) and has been deemed a poor disperser.

The low levels of genetic differentiation identified between populations from the five independent rivers on Hainan Island suggested that the Hainan Island populations might constitute a single panmictic population, even though these rivers have been thought to be distinct river systems separated by known barriers, such as the Yinggeling and Wuzhishan Mountains [9,47]. Our results were incongruent with previous studies of *G. orientalis* and *Channa gachua*, both which recovered deeply divergent lineages among the Hainan Island river populations [9,47]. This discrepancy is likely due to mobility differences between *T. houdemeri* and the other two species. *T. houdemeri* is a mesopelagic fish, while *G. orientalis* and *C. gachua* are benthic species [14,48]. Therefore, *T. houdemeri* may have relatively higher dispersal potential than either *G. orientalis* or *C. gachua*. Indeed, the five rivers are spatially close (on the scale of tens of kilometers), and some river sections may occasionally be physically connected by seasonal flooding. Consistently, previous studies have suggested that seasonal flooding may promote migration in some fish taxa, such as *Arapaima gigas* [49] and the genus *Marcopodus* [50]. Moreover, population migration via the continental shelf during more recent glacial periods is a non-negligible factor.

### 4.2. Demographic History

Neutrality tests, mismatch distribution analyses, and EBSP analyses supported the recent population expansion of *T. houdemeri*. EBSP analyses suggested the expansion event had occurred ~0.060–0.120 Ma, during the Late Pleistocene. Geological reports have argued that southern China experienced an interglacial period during the Late Pleistocene (0.126–0.018 Ma) [42]. During interglacial periods, the warm climate may have promoted the expansion of *T. houdemeri* populations, as has previously been proposed in *C. recurviceps* [8]. Taken together, our results indicated that climate fluctuations during the Late Pleistocene shaped the demographic history of *T. houdemeri* populations. However, we found that *T. houdemeri* populations remained stable during the Last Glacial Maximum (~0.021 Ma), suggesting that the Last Glacial Maximum did not impact the demography of this species. The regions in southern China are temperate and tropical zones, which might be less influenced by the Last Glacial Maximum. Similar patterns were also reported in other fish species in southern China [49,51,52], such as *Hemiculter leucisculus* [51] and *Ochetobius elongatus* [52].

### 4.3. Genetic Diversity

High levels of H*d* and low levels of π observed in *T. houdemeri* populations using both MCR and *RAG2* were indicative of a population bottleneck followed by rapid population expansion [53]. A signal of recent population expansion occurred in *T. houdemeri* populations was also demonstrated by different analytical methods. The global genetic diversity level using concatenated *Cytb* and CR of *T. houdemeri* populations was slightly lower than that of *M. terminalis* populations [7]. However, the genetic diversity level using *RAG2* was higher than that of *M. terminalis* populations [7]. Furthermore, our study uncovered the obvious difference of genetic diversity level between the four genetic groups (Table 1). One possible reason was that the *T. houdemeri* members within the four genetic groups were derived from the ancestral populations through a recent population expansion.

### 4.4. Conservation Implications

Delineating the population structures of widespread taxa remains inherently difficult, as many populations of widespread species have dwindled or even disappeared in certain sections of their distribution ranges due to human interference and/or environmental degradation [54]. It is essential to identify population boundaries to support conservation and management programs aiming to ensure the continued persistence of such species. It was clear that the *T. houdemeri* populations were comprised of four mitochondrial haplotype groups that were distributed in separate river systems. The genetic endemism uncovered in these independent rivers deserves specific consideration. Special attention should also be paid to the Moyang River, the Jian River, and the Lian River because these small rivers harbor two unique mitochondrial haplotype groups. The low genetic diversity and fragmented distributions of *T. houdemeri* in these three rivers increase the susceptibility of these populations to adverse effects, such as inbreeding depression and diseases, leading to a high threat of extinction [55,56]. Importantly, Hainan Island populations and mainland populations should be treated as two independent management units (MUs [57]) due to their obvious population boundary and high level of genetic differentiation.

## 5. Conclusions

Our study provides the first comprehensive view of the genetic structure *T. houdemeri* populations based on wide geographic sampling and both mitochondrial and nuclear markers. Genetic endemism with clear geographic boundaries was identified in the *T. houdemeri* populations. River landscape changes during the Late Pleistocene, independent river systems, biogeographic barriers, and dispersal potential have played key roles in shaping the genetic structure and demographic dynamics of *T. houdemeri*. The results of this study may help improve monitoring and protection efforts targeting *T. houdemeri*.

## Figures and Tables

**Figure 1 biology-11-01641-f001:**
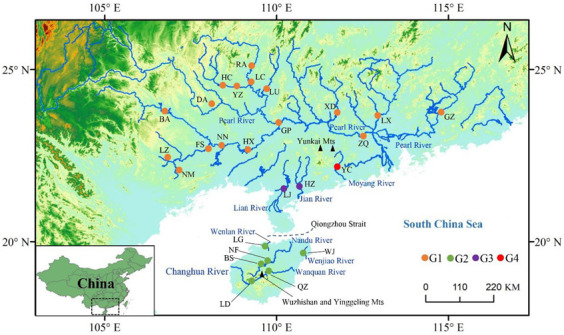
*Toxabramis houdemeri* sampling sites in the current study. The pie charts represent the four mitochondrial groups defined based on the mitochondrial network. Population codes are defined in Table 1.

**Figure 2 biology-11-01641-f002:**
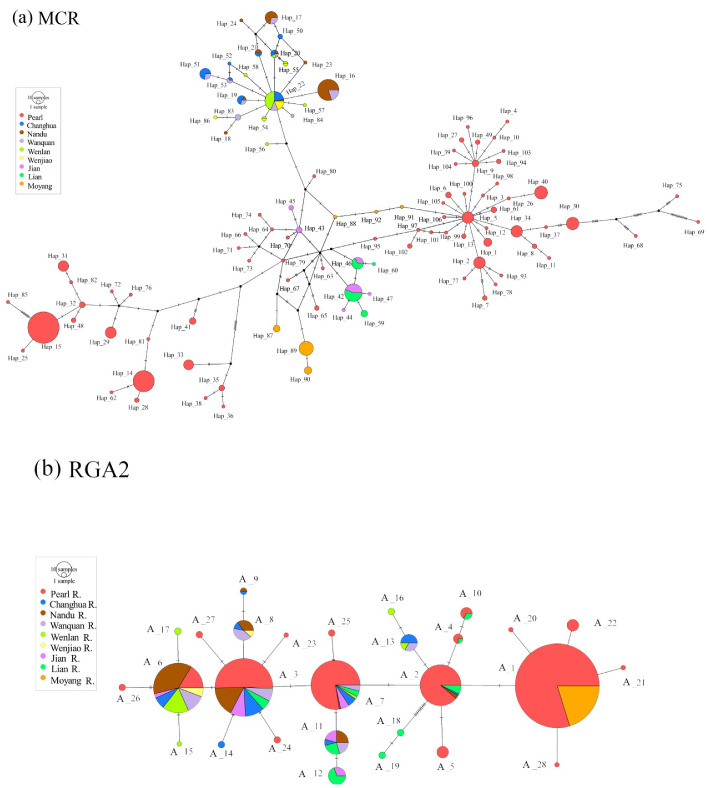
Haplotype (**a**) and allele (**b**) networks for the nine rivers included in this study based on concatenated mitochondrial locus (MCR) and *RAG2* data. Circles are proportional to sample size (scale differs between networks) and colored based on geographical origin.

**Figure 3 biology-11-01641-f003:**
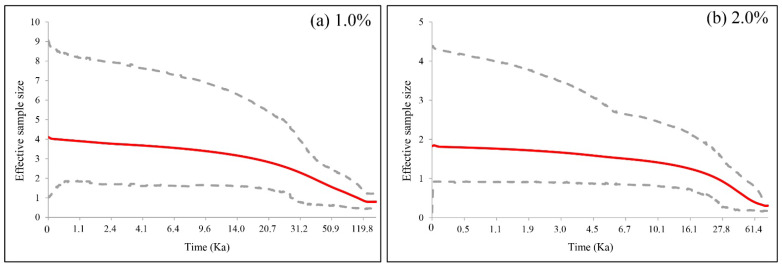
Extended Bayesian skyline plots of the *Cytb* gene using substitution rates of 1.0% (**a**) and 2.0% (**b**) per million years (Myr). The ordinate corresponds to Neτ, the product of effective population size and generation length per millennium (ka), and the abscissa corresponds to time (ka). Estimates of means are joined by a red line, while the dashed lines delineate the 95% HPD limits.

**Figure 4 biology-11-01641-f004:**
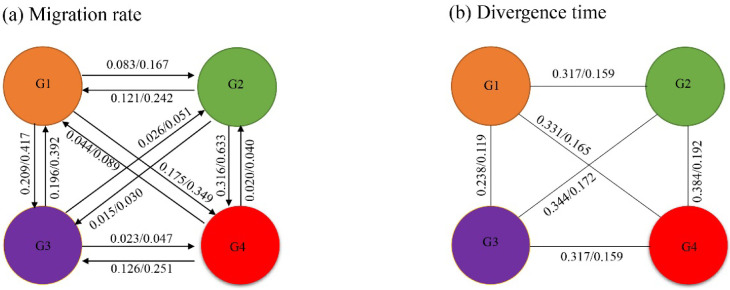
Historical immigration rates (**a**) and divergence times (**b**) between the four defined mitochondrial haplotype groups (G1–G4) as derived from the mitochondrial network. Immigration rates (**a**) and divergence times (**b**) were inferred based on concatenated mitochondrial loci (MCR) with a substitution rates of 0.756% (left) and 1.513% (right).

**Table 1 biology-11-01641-t001:** Genetic diversity indexes for *Toxabramis houdemeri* populations based on concatenated mitochondrial locus (MCR) and *RAG2*. *n*, sequence numbers; H, haplotype numbers; H*d*, haplotype diversity; and *π*, nucleotide diversity.

Locations	Codes	River	MCR				RAG2			
			*n*	H	H*d*	*π*	*n*	H	H*d*	*π*
Baise	BA	Pearl R.	28	3	0.611 ± 0.060	0.0029 ± 0.0003	56	20	0.933 ± 0.015	0.0027 ± 0.0002
Duan	DA	Pearl R.	2	1	—	—	2	1	—	—
Fushui	FS	Pearl R.	22	2	0.091 ± 0.081	0.0001 ± 0.0001	44	9	0.796 ± 0.049	0.0023 ± 0.0001
Guiping	GP	Pearl R.	26	9	0.868 ± 0.040	0.0035 ± 0.0005	52	15	0.880 ± 0.033	0.0027 ± 0.0002
Guzhu	GZ	Pearl R.	33	8	0.782 ± 0.045	0.0056 ± 0.0003	66	7	0.304 ± 0.073	0.0005 ± 0.0001
Hechi	HC	Pearl R.	11	1	0	0	22	9	0.896 ± 0.034	0.0018 ± 0.0001
Hengxian	HX	Pearl R.	6	4	0.867 ± 0.129	0.0041 ± 0.0008	12	5	0.848 ± 0.067	0.0032 ± 0.0003
Liucheng	LC	Pearl R.	19	8	0.830 ± 0.066	0.0040 ± 0.0004	38	13	0.902 ± 0.022	0.0024 ± 0.0001
Lixi	LX	Pearl R.	28	13	0.905 ± 0.034	0.0013 ± 0.0002	48	20	0.855 ± 0.043	0.0023 ± 0.0003
Longzhou	LZ	Pearl R.	26	18	0.954 ± 0.027	0.0059 ± 0.0011	52	21	0.956 ± 0.013	0.0029 ± 0.0002
Luzhai	LU	Pearl R.	11	8	0.927 ± 0.066	0.0047 ± 0.0004	22	13	0.952 ± 0.024	0.0026 ± 0.0002
Nanning	NN	Pearl R.	7	7	1.0 ± 0.076	0.0028 ± 0.0005	14	6	0.879 ± 0.052	0.0034 ± 0.0004
Ningming	NM	Pearl R.	17	4	0.331 ± 0.143	0.0017 ± 0.0007	30	9	0.703 ± 0.085	0.0015 ± 0.0002
Rongan	RA	Pearl R.	28	2	0.071 ± 0.065	0.0003 ± 0.0003	56	14	0.647 ± 0.073	0.0018 ± 0.0003
Xindu	XD	Pearl R.	1	1	—	—	2	2	—	—
Yizhou	YZ	Pearl R.	17	2	0.309 ± 0.122	0.0021 ± 0.0009	34	7	0.793 ± 0.040	0.0018 ± 0.0002
Zhaoqing	ZQ	Pearl R.	30	22	0.972 ± 0.017	0.0033 ± 0.0004	50	15	0.788 ± 0.054	0.0020 ± 0.0003
Baisha	BS	Nandu R.	30	9	0.632 ± 0.096	0.0010 ± 0.0002	60	15	0.811 ± 0.035	0.0014 ± 0.0001
Nanfeng	NF	Nandu R.	20	2	0.442 ± 0.087	0.0009 ± 0.0002	40	12	0.897 ± 0.025	0.0023 ± 0.0001
Ledong	LD	Changhua R.	27	8	0.835 ± 0.043	0.0010 ± 0.0010	54	24	0.942 ± 0.016	0.0024 ± 0.0002
Lingao	LG	Wenlan R.	20	7	0.521 ± 0.135	0.0005 ± 0.0002	30	9	0.821 ± 0.048	0.0023 ± 0.0004
Qiongzhong	QZ	Wanquan R.	28	9	0.873 ± 0.037	0.0011 ± 0.0001	56	23	0.944 ± 0.015	0.0025 ± 0.0002
Wenjiao	WJ	Wenjiao R.	9	4	0.583 ± 0.183	0.0007 ± 0.0003	14	7	0.890 ± 0.055	0.0021 ± 0.0002
Huazhou	HZ	Jian R.	23	6	0.700 ± 0.088	0.0008 ± 0.0002	44	14	0.895 ± 0.03-	0.0018 ± 0.0001
Lianjiang	LJ	Lian R.	28	4	0.632 ± 0.067	0.0004 ± 0.0001	60	30	0.953 ± 0.014	0.0047 ± 0.0008
Yangchun	YC	Moyang R.	30	6	0.611 ± 0.088	0.0013 ± 0.0003	60	1	0	0
G1		—	312	71	0.895 ± 0.013	0.0044 ± 0.0002	600	91	0.873 ± 0.012	0.0027 ± 0.0001
G2		—	134	21	0.835 ± 0.020	0.0010 ± 0.0001	254	56	0.912 ± 0.011	0.0023 ± 0.0001
G3		—	51	8	0.665 ± 0.058	0.0006 ± 0.0001	104	38	0.935 ± 0.014	0.0036 ± 0.0005
G4		—	30	6	0.611 ± 0.088	0.0013 ± 0.0003	60	1	0	0
Global		—	527	106	0.948 ± 0.0049	0.0044 ± 0.0001	1018	168	0.915 ± 0.0064	0.0036 ± 0.0001

**Table 2 biology-11-01641-t002:** Hierarchical population structure of *Toxabramis houdemeri* populations based on concatenated mitochondrial locus (MCR) and *RAG2*.

Source of Variation	MCR			RAG2		
	Percentage of Variation	Φ-Statistic	*p*-Value	Percentage of Variation	Φ-Statistic	*p*-Value
Grouped by locations						
Between populations	53.94	0.539	<0.001	41.44	0.414	<0.001
Within populations	46.06	0.461		58.56	0.586	
Grouped by nine rivers						
Between rivers	25.26	0.253	<0.001	32.15	0.322	<0.001
Between populations within rivers	32.65	0.327	<0.001	15.69	0.157	<0.001
Within populations	42.09	0.421	<0.001	52.16	0.522	<0.001
Grouped by genetic groups						
Between genetic groups	37.98	0.379	<0.001	37.74	0.377	<0.001
Between populations within genetic groups	22.91	0.229	<0.001	12.40	0.124	<0.001
Within populations	39.11	0.391	<0.001	49.86	0.499	<0.001

## Data Availability

DNA sequences have been deposited in GenBank under Accession numbers Details regarding individual samples are available in Appendix A.

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
