# Peer review of "Genetic Structure of an East Asian Minnow (Toxabramis houdemeri) in Southern China, with Implications for Conservation"

_biology, 2022, doi:10.3390/biology11111641_

Round 1

Reviewer 1 Report

The authors present an interesting manuscript on genetic structure and phylogeographic patterns of Toxabramis houdemeri. Study design and research questions are clearly described. In this sense, it is easy to understand the aim of this study. The bright side of the manuscript is that to provide some useful practical details on related topic. In this context, the study contributes to conservation and understanding of genetic structure and phylogeographic patterns of Toxabramis houdemeri. However, there are some missing points in the manuscripts. Therefore, I would like to make some suggestions to improve the quality of the paper as below:

Authors mentioned “genetic endemism” in abstract and introduction”. This term should be clearly explained with related papers and difference from evolutionary significant unit and management unit in the introduction section. Did authors have the samples from all Toxabramis houdemeri populations?

Some sentences have a grey background (e.g. lines 7-9, 46, 57-58, 74-75, 149-150, 158-162). Please delete grey backgrounds.

Line 60: structures -> structure

Line 128: [CITE] ? Please put a reference here.

Line 199: The resolution of Figure is low. A high-resolution figure would be better. Also, I think, larger figure would be better since it is not easy to see haplotype numbers in current figure. Figure a may on the top and figure b may be below the figure a.

Line 287: “endemic mitochondrial haplotypes” What is the difference of these haplotypes from evolutionary significant units and management units?

Lines 298-299: private mitochondrial haplotypes -> unique mitochondrial haplotypes

Table captions should be added to Supporting information files.

Author Response

The authors present an interesting manuscript on genetic structure and phylogeographic patterns of Toxabramis houdemeri. Study design and research questions are clearly described. In this sense, it is easy to understand the aim of this study. The bright side of the manuscript is that to provide some useful practical details on related topic. In this context, the study contributes to conservation and understanding of genetic structure and phylogeographic patterns of Toxabramis houdemeri. However, there are some missing points in the manuscripts. Therefore, I would like to make some suggestions to improve the quality of the paper as below:

Reply: Thanks for your valuable suggestions and comments. We will carefully revise this draft following your suggestions.

Authors mentioned “genetic endemism” in abstract and introduction”. This term should be clearly explained with related papers and difference from evolutionary significant unit and management unit in the introduction section. Did authors have the samples from all Toxabramis houdemeri populations?

Reply: First, we have added explanation in the Introduction section (lines 46-7, page 2). Second, past literatures showed Toxabramis houdemeri is endemic in southern China and widely resides in southern China drainages. In this study, our samples have covered its most distribution ranges including many independent drainage systems.

Some sentences have a grey background (e.g. lines 7-9, 46, 57-58, 74-75, 149-150, 158-162). Please delete grey backgrounds.

Reply: We have checked and revised these grey backgrounds in the revised draft.

Line 60: structures -> structure

Reply: Done. (line 59, page 2)

Line 128: [CITE] ? Please put a reference here.

Reply: Added. (line 59, page 4)

Line 199: The resolution of Figure is low. A high-resolution figure would be better. Also, I think, larger figure would be better since it is not easy to see haplotype numbers in current figure. Figure a may on the top and figure b may be below the figure a.

Reply: Done. We have enlarged the figure and kept 300 dpi resolution ratio.

Line 287: “endemic mitochondrial haplotypes” What is the difference of these haplotypes from evolutionary significant units and management units?

Reply: Endemic mitochondrial haplotypes mean private haplotypes in particular geographic regions, while the haplotypes from evolutionary significant units and management units should own markable phylogenetic boundary between and/or among compared geographic regions. Endemic mitochondrial haplotypes often mixed with haplotypes from other geographic regions without haplotype sharing, while haplotypes from different evolutionary significant units and management units indicate they belong to different genetic groups with genetic and geographic boundary.

Lines 298-299: private mitochondrial haplotypes -> unique mitochondrial haplotypes

Reply: Revised.(line 297, page 9)

Table captions should be added to Supporting information files.

Reply: Done. (see supporting tables)

Reviewer 2 Report

This manuscript entitled "Phylogeographic patterns of an East Asian minnow (Toxabramis houdemeri) in southern China, with implications for conservation" analyzed the population genetic structures and phylogeographic patterns of an East Asian minnow by using mtDNA and nuDNA markers. The strength of the study is that have a large sampling group.

There are some comments as follows:

1, The phylogeographic patterns of T. houdemer wasn’t clear. Did the four groups belong to different lineages? The authors needed to construct phylogenetic tree. If there were four different lineages, the authors should conduct demographic analyses in different lineages, including the results of the mismatch distribution analyses as graphs.

2, Please added the table captions in supplementary information.

3, Was the significance adjusted using sequential Bonferroni corrections in Table S5?

4, The genetic diversity level among the four groups showed in Table 1 showed obvious difference, why?

5, Line 223, though the p value was significant, the values of r in the mantel test were very low, I thought that the T. houdemeri was not a good fit for the isolation-by-distance pattern.

5, In 4.2 Demographic history part, the authors discussed the effects of interglacial period during the Late Pleistocene, what about last glacial maximum?

6, An overall suggestion is that, there are a few grammar errors in the main text, and the language need to be further polished.

Author Response

Comments and Suggestions for Authors

This manuscript entitled "Phylogeographic patterns of an East Asian minnow (Toxabramis houdemeri) in southern China, with implications for conservation" analyzed the population genetic structures and phylogeographic patterns of an East Asian minnow by using mtDNA and nuDNA markers. The strength of the study is that have a large sampling group.

Reply: Thanks for your valuable suggestions and comments. We will revise the draft following your advice.

There are some comments as follows:

1, The phylogeographic patterns of T. houdemer wasn’t clear. Did the four groups belong to different lineages? The authors needed to construct phylogenetic tree. If there were four different lineages, the authors should conduct demographic analyses in different lineages, including the results of the mismatch distribution analyses as graphs.

Reply: Thanks for your suggestions. We have conducted Bayesian and maximum likelihood trees and we have not detected robust pattern of lineage split (see below). In addition, the mean k2p distance within the overall populations using MCR sequences is relatively low (0.0044) and so we only built network instead. Therefore, we have conducted demographic analyses aiming to the overall populations.

2, Please added the table captions in supplementary information.

Reply: Done.(see supporting tables)

3, Was the significance adjusted using sequential Bonferroni corrections in Table S5?

Reply: Yes. We have added the note in the caption of Table S4 and S5.

4, The genetic diversity level among the four groups showed in Table 1 showed obvious difference, why?

Reply: Apart from Group 1, the remaining three genetic groups distributed in limited regions and their population size might be lower than that in Group 1, which occupied larger distribution ranges. Furthermore, different sample frequencies and sample size in some populations would omit some genetic diversity. The abovementioned reasons may cause the different level of genetic diversity.

5, Line 223, though the p value was significant, the values of r in the mantel test were very low, I thought that the T. houdemeri was not a good fit for the isolation-by-distance pattern.

Reply: We have re-written this viewpoint in the revised draft. (lines 223-4, page 7)

5, In 4.2 Demographic history part, the authors discussed the effects of interglacial period during the Late Pleistocene, what about last glacial maximum?

Reply: We have added some contents in the demographic history part regarding the LGM. (lines 341, page 345)

6, An overall suggestion is that, there are a few grammar errors in the main text, and the language need to be further polished.

Reply: We have checked overall MS and conducted language editing by LETPUB COMPANY.

Round 2

Reviewer 2 Report

I was disappointed that I didn’t see any polish traces in the revised manuscript.

The authors didn’t detect robust pattern of lineage split, which indicated that there was only one lineage in southern China. Phylogeography deals with the spatial arrangements of genetic lineages, so I think that the title involved of “phylogeographic patterns” was not appropriate.

About the obvious difference of genetic diversity level among the four groups, I think one possible reason is that the genetic diversity of the ancestral population was higher than those of the populations which undergone a recent population expansion. And please add the result of mismatch analysis. And this part should be discussed in the discussion part. 

The author agrees that the T. houdemeri was not a good fit for the isolation-by-distance pattern. So please rewrite the sentences in line 315-316.

The author thought that the Last Gacial Maximum did not significantly impact the demography of this species, is there any other species which showed similar pattern? Please added some references.

Author Response

Comments and Suggestions for Authors

I was disappointed that I didn’t see any polish traces in the revised manuscript.

Reply:We have resubmitted to Letpub Company for second round English editing and highlight with blue in the revised draft.

The authors didn’t detect robust pattern of lineage split, which indicated that there was only one lineage in southern China. Phylogeography deals with the spatial arrangements of genetic lineages, so I think that the title involved of “phylogeographic patterns” was not appropriate.

Reply: Done. We have revised the title as ‘Genetic structure of an East Asian minnow (Toxabramis houdemeri) in southern China, with implications for conservation’.

About the obvious difference of genetic diversity level among the four groups, I think one possible reason is that the genetic diversity of the ancestral population was higher than those of the populations which undergone a recent population expansion. And please add the result of mismatch analysis. And this part should be discussed in the discussion part. 

Reply: Done. We have added mismatch analyses in the revised draft. (Lines 160-162, page 5; lines 237-241, page 7; page 338, page 10).

The author agrees that the T. houdemeri was not a good fit for the isolation-by-distance pattern. So please rewrite the sentences in line 315-316.

Reply:Done. We have deleted the sentence in the revised draft. (lines 319-320, page 10)

The author thought that the Last Gacial Maximum did not significantly impact the demography of this species, is there any other species which showed similar pattern? Please added some references.

Reply: We have added some similar pattern that occurred in fish species in the discussion part. (lines 345-351, page 11)

Round 3

Reviewer 2 Report

 The author has revised the title as ‘Genetic structure of an East Asian minnow (Toxabramis houdemeri) in southern China, so please rewritten the sentences involved of “phylogeographic patterns” in the manuscript.

 The author didn’t reply to my question about genetic diversity:About the obvious difference of genetic diversity level among the four groups, I think one possible reason is that the genetic diversity of the ancestral population was higher than those of the populations which undergone a recent population expansion. ” Furthermore, I think that it rarely makes sense to list the values of genetic diversity simply in the line 226-234 without comparison and analysis. Is the genetic diversity level of Toxabramis houdemeri among the four groups high or low compared with other freshwater fishes in the southern China.

Author Response

 The author has revised the title as ‘Genetic structure of an East Asian minnow (Toxabramis houdemeri) in southern China, so please rewritten the sentences involved of “phylogeographic patterns” in the manuscript.

Reply: We have rewritten the sentences involved of “phylogeographic patterns” in the revised manuscript.

 The author didn’t reply to my question about genetic diversity:“ About the obvious difference of genetic diversity level among the four groups, I think one possible reason is that the genetic diversity of the ancestral population was higher than those of the populations which undergone a recent population expansion. ” Furthermore, I think that it rarely makes sense to list the values of genetic diversity simply in the line 226-234 without comparison and analysis. Is the genetic diversity level of Toxabramis houdemeri among the four groups high or low compared with other freshwater fishes in the southern China.

Reply: We are sorry to forget to reply this question. In the revised ms, we have added this content in the discussion section as ‘4.3 Genetic diversity’(lines 349-360, page 11). Furthermore, we have rewritten the description of the genetic diversity indexes (lines 225-230, page 7). Because different genes and combined number of gene will influence genetic diversity calculation. Furthermore, given that different genetic groups belong to rigid geographic regions, it is not suitable to compared the genetic diversity level among the four groups with an independent species. Therefore, we tried to compared the genetic diversity level of global Toxabramis houdemeri populations with a Cultrinae species in southern China, named as Megalobrama terminalis(lines 349-360, page 11). The compared study employed same gene to clarify the genetic diversity pattern.
